# Trans-Regulation of Alternative *PD-L1* mRNA Processing by CDK12 in Non-Small-Cell Lung Cancer Cells

**DOI:** 10.3390/cells12242844

**Published:** 2023-12-15

**Authors:** Trine V. Larsen, Christoffer T. Maansson, Tina F. Daugaard, Brage S. Andresen, Boe S. Sorensen, Anders L. Nielsen

**Affiliations:** 1Department of Biomedicine, Aarhus University, 8000 Aarhus, Denmark; tvl@biomed.au.dk (T.V.L.); ctm@clin.au.dk (C.T.M.); tfm@biomed.au.dk (T.F.D.); 2Department of Clinical Medicine, Aarhus University, 8200 Aarhus, Denmark; boesoere@rm.dk; 3Department of Clinical Biochemistry, Aarhus University Hospital, 8200 Aarhus, Denmark; 4Department of Biology and Molecular Biology, Southern University of Denmark, 5230 Odense, Denmark; bragea@bmb.sdu.dk

**Keywords:** immunotherapy, lung cancer, PD-L1, splicing, IFN-γ, CDK12

## Abstract

Immunotherapy using checkpoint inhibitors targeting the interaction between PD-1 on T cells and PD-L1 on cancer cells has shown significant results in non-small-cell lung cancer (NSCLC). Not all patients respond to the therapy, and PD-L1 expression heterogeneity is proposed to be one determinant for this. The alternative processing of *PD-L1* RNA, which depends on an alternative poly-A site in intron 4, generates a shorter mRNA variant (*PD-L1v4*) encoding soluble PD-L1 (sPD-L1), relative to the canonical *PD-L1v1* mRNA encoding membrane-associated PD-L1 (mPD-L1). This study aimed to identify factors influencing the ratio between these two *PD-L1* mRNAs in NSCLC cells. First, we verified the existence of the alternative *PD-L1* RNA processing in NSCLC cells, and from in silico analyses, we identified a candidate list of regulatory factors. Examining selected candidates showed that CRISPR/Cas9-generated loss-of-function mutations in *CDK12* increased the *PD-L1v4*/*PD-L1v1* mRNA ratio and, accordingly, the sPD-L1/mPD-L1 balance. The CDK12/13 inhibitor THZ531 could also increase the *PD-L1v4*/*PD-L1v1* mRNA ratio and impact the *PD-L1* transcriptional response to IFN-γ stimulation. The fact that CDK12 regulates *PD-L1* transcript variant formation in NSCLC cells is consistent with CDK12’s role in promoting transcriptional elongation over intron-located poly-A sites. This study lays the groundwork for clinical investigations to delineate the implications of the CDK12-mediated balancing of sPD-L1 relative to mPD-L1 for immunotherapeutic responses in NSCLC.

## 1. Introduction

Non-small-cell lung cancer (NSCLC) is a leading cause of cancer-related death. NSCLC is a heterogeneous disease and can be associated with oncogenic driver mutations activating, e.g., Epidermal Growth Factor Receptor (EGFR), Kirsten Rat Sarcoma Virus (KRAS), and Anaplastic Lymphoma Kinase (ALK) [1]. Chemotherapy and targeted therapies addressing specific oncogenic drivers are treatment options, but the initial clinical effect is followed by treatment resistance [1]. Immunotherapy is an additional treatment option for NSCLC, either alone or in combination with other therapies [2,3,4,5,6,7,8,9]. This is illustrated with Programmed Cell Death Protein 1 (PD-1)/Programmed Death-Ligand 1 (PD-L1) immunotherapy, where therapeutic antibodies are used to block the interaction between PD-1 and PD-L1, thereby abrogating the PD-1/PD-Ligand axis involved in peripheral and central immune cell tolerance [10]. PD-1 is a co-inhibitory surface receptor expressed on a wide range of immune cells, including antigen-stimulated T cells [11,12]. PD-L1 belongs to the immunoglobulin (Ig) superfamily, and PD-L1 and the homologous PD-L2 are together ligands for PD-1 [13,14]. Cancer and stromal cells can express PD-L1, and oncogenic driver mutations stimulate the expression of PD-L1 [15,16,17,18,19]. Moreover, immune-cell-produced pro-inflammatory molecules, such as interferons (IFNs) of types I (IFN-α and IFN-β), II (IFN-γ), and III (IFN-λ), also induce cancer cell PD-L1 expression [20,21]. The interaction of PD-1 with PD-L1 alters T cell activity, including the inhibition of T cell proliferation, survival, and cytokine/IFN production [15]. The result of PD-L1 production by cancer cells is the creation of an immunosuppressive tumor microenvironment that is beneficial for the proliferation and survival of tumor cells. The response rate to PD-1/PD-L1 immunotherapy is approximately 20%, illustrating the challenge from therapy resistance, either innate or acquired [5,7,22,23]. The quantification of membrane-bound PD-L1 (mPD-L1) is currently used as a predictive biomarker for an NSCLC immunotherapy response [7,8,9,24,25,26,27]. However, around 15% of patients scored with PD-L1-negative tumors will respond, and many patients with tumors scored with a positive PD-L1 TPS do not experience the benefits [3,9,28]. Secreted PD-L1 (secPD-L1), either soluble (sPD-L1) or presented by exosomes (exoPD-L1), is recently shown to have prognostic and predictive importance for NSCLC immunotherapy [29,30,31,32]. sPD-L1 can be produced by two non-exclusive pathways, the proteolytic processing of mPD-L1/exoPD-L1 and the alternative processing of *PD-L1* mRNA transcribed from the cluster of differentiation 274 (CD274) gene (also abbreviated as *PD-L1* gene) to encode PD-L1 isoforms lacking the transmembrane domain [29]. For the latter, of the different alternative *PD-L1* mRNA variants encoding sPD-L1, the most common is *PD-L1v4* (NM_001314029) [33,34,35,36,37,38,39]. Canonical *PD-L1* mRNA, *PD-L1v1*, includes seven exons, whereas *PD-L1v4* mRNA is a result of the skipped use of the canonical exon 4 donor splice site, together with the use of an otherwise skipped polyadenylation site contributed from an integrated long interspersed nuclear element (LINE), the *L2A* element, located downstream in intron 4 [34,35,36]. A consequence of the novel 3′-end sequence in *PD-L1v4* mRNA is the production of the PD-L1v4 isoform, which retains the normal PD-L1 leader peptide and extracellular Ig domains but includes a novel 18 amino acid C-terminal sequence, alternatively to the trans-membrane and intracellular domains in mPD-L1 [34,35,36,37,38]. PD-L1v4 is potentially N-linked glycosylated in the unique C-terminal, which also supports PD-L1v4 covalent homodimer and higher-order multimer formations [35,36]. In silico analyses addressing *PD-L1v1* mRNA and *PD-L1v4* mRNA expression in The Cancer Genome Atlas (TCGA) and the Cancer Cell Line Encyclopedia (CCLE) datasets indicated that *PD-L1v4* expression, in general, is lower (by around 10-fold) than for *PD-L1v1* mRNA [34,35,36,37]. Moreover, the ratio of *PD-L1v4* relative to *PD-L1v1* mRNA is increased in cancer tissue relative to normal tissue [35,36]. Finally, the expression of *PD-L1v4* and *PD-L1v1* mRNA is more correlated in cancer cells than is observed in normal tissue; however, the two mRNA variants are regulated differently as illustrated, e.g., by the presence of non-synchronous induction patterns following IFN stimulation [34,35,36,37]. PD-L1v4 can bind PD-1, but the impact of PD-L1v4 on immunotherapy is diversely described [34,35,36,37,38]. Whereas some studies have identified that PD-L1v4 has a direct suppressive function for activated T cells, e.g., mediating a reduced secretion of IFN-γ [34,35,38], other studies addressing more physiological amounts of PD-L1v4 have not been able to identify this [36,37]. In these studies, PD-L1v4 functions in immunotherapy were defined to be either acting as a PD-1 receptor antagonist, competing for the immunosuppressive function of mPD-L1, or being a decoy for anti-PD-L1 antibodies [36,37]. The latter functionality has been identified in general for sPD-L1, and could contribute to resistance towards PD-L1 immunotherapy [33,37].

The expression of *PD-L1v4* mRNA in cancer, including NSCLC, and the functional impact of the derived sPD-L1 protein for cancer immunotherapy has been described [34,35,36,37,38]. However, details of the trans-regulatory mechanisms regulating the *PD-L1v4*/*PD-L1v1* mRNA ratio, and accordingly, their important contribution when defining the balance of sPD-L1 and mPD-L1 in the tumor microenvironment, are not yet revealed. In this study, we searched for trans-regulatory RNA-processing mechanisms regulating the *PD-L1* mRNA variant ratio, and identified the RNA polymerase II (POLII) elongation and processivity factor CDK12 to be such a regulator.

## 2. Materials and Methods

### 2.1. Cell Culture

Details concerning the used NSCLC cell lines, including the site of purchase and growth conditions, were described in [40]. For IFN induction, the cells were stimulated with 10 ng/mL of IFN-γ (PeproTech, London, UK, 300-02) for 24 h. For the control treatment, the cells received an equal volume of PBS + 0.01% bovine serum albumin (Sigma-Aldrich, Saint Louis, MO, USA, A2153). For THZ531 treatment, the cells were given the indicated concentrations of THZ531 (Sigma-Aldrich, Saint Louis, MO, USA, SML2619) in solvent, DMSO, for 24 h, or equal concentrations of DMSO for control.

### 2.2. Tissue from NSCLC Tumors

The expression of *PD-L1* mRNA variants was examined in lung tissues from NSCLC adenocarcinoma tumors (*n* = 31) removed via resection or lobectomy at Aarhus University Hospital during the diagnostics and treatment of the tumors. Tissue preparation and the selection of tissue was performed as previously described [40]. The use of anonymized lung tissue from tumors with NSCLC was requested and approved according to Danish and EU ethical guidelines.

### 2.3. RNA Extraction, cDNA Synthesis, Reverse Transcriptase Quantitative PCR (RT-qPCR), and Droplet Digital PCR (ddPCR)

RNA was extracted using the Tri Reagent (Sigma-Aldrich, Saint Louis, MO, USA, T9424), and after DNAse treatment, cDNA synthesis from 1 μg of RNA was performed with an iScript cDNA Synthesis Kit (Bio-Rad, Hercules, CA, USA, 170-8890). The ddPCR detection of *PD-L1* variants was performed on 60 ng of cDNA, and the detection of *TBP* was performed on 30 ng of cDNA in each reaction. The ddPCR was performed essentially as described in [41], but here targeting *PD-L1* mRNA species. Briefly, droplets were made using the QX200 AutoDG (Bio-Rad, Hercules, CA, USA). Semi-skirted ddPCR plates (Bio-Rad, Hercules, CA, USA) were used, and ddPCR was performed using a GeneAmp PCR System 9700 (Applied Biosystems, Waltham, MA, USA). Each sample was run in three technical replicates at 95 °C for 10 min, with 40 cycles of PCR amplification (94 °C for 30 s, primer-specific annealing temperature for 1 min), and 98 °C for 10 min. The droplets were read using the QX200 Droplet Reader (Bio-Rad, Hercules, CA, USA). The primer and probe sequences are given in Appendix A. The concentration for probes was 250 nM, and for forward and reverse primers, 450 nM. The calculation of copies/uL was carried out automatically in QX Manager 1.2 Standard Edition (Bio-rad, Hercules, CA, USA) using the Poisson error model with 95% confidence intervals. Excel was used to convert this into copies/ug of RNA in the starting material. RT-qPCR was performed as previously described [21,40], and the used primers are shown in Appendix A.

### 2.4. CRISPR/Cas9 Procedures

sgRNAs were designed using the UCSC Genome Browser. For the control, a scrambled sgRNA, sgRNA C, with no target homology in the human genome was used. The sgRNA sequences are shown in Appendix A. Potential sgRNA off-targets were determined using the Clustered Regularly Interspaced Short Palindromic Repeats (CRISPR) Targets tool on the UCSC Genome Browser. Only sgRNAs with potential off-targets containing a minimum of 2 mismatches were used. Furthermore, the MIT specificity score and the Cutting Frequency Determination (CFD) score were used to select sgRNAs with few potential off-targets. The potential off-targets with the highest CFD scores needed to be located in intergenic or intron regions for a sgRNA to be selected. The protocol used to obtain CRISPR/Cas9-mediated genetic modifications was previously described [42]. Briefly, sgRNA oligonucleotides were cloned in pLentiGuide-Puromycin (Addgene, Watertown, MA, USA, Cat#52963) or pLentiGuide-Hygromycin (Addgene, Watertown, MA, USA, Cat#139462). The correct insertion of the sgRNA oligonucleotides was verified via sequencing, using a primer pair targeting the U6 promoter. Genetic modifications in HCC827Cas9 cells were performed using the lentiviral transduction of sgRNA-expressing vectors, and cells were allowed growth for a minimum of 14 days with hygromycin or puromycin resistance selection. For the analysis of insertion–deletion (indel) and knock-out efficiency, genomic DNA was extracted using an EZNA tissue DNA kit (Omega Biotek, Norcross, GA, USA). Genomic regions of interest were amplified using 50 ng of DNA and HotStarTaq polymerase (Qiagen, Venlo, The Netherlands) or Taq DNA Polymerase Master Mix RED (AmpliQon, Odense, Denmark). The DNA sequences of the PCR products were analyzed for indel and knock-out scores using the Synthego ICE analysis tool (https://labs.synthego.com/) (accessed on 1 September 2023). We analyzed cell populations instead of cell clones to minimize phenotype effects arising from the outgrowth of colonies from single cells.

### 2.5. Western Blotting and Enzyme-Linked Immunosorbent Assay (ELISA)

The procedures for Western blotting and the used antibodies were described previously [40,43]. An ELISA with the Human PD-L1 ELISA Kit (Abcam, Cambridge, UK, ab277712) was carried out according to the recommendations from the supplier, except that incubation with the assay development solution was performed for 15 min. A standard curve was created using a four-parameter curve fit, and the PD-L1 concentration was determined from the standard curve.

### 2.6. In Silico Analysis and Statistics

The transcripts per million (TPM) values of a subset of lung cancer cell lines were collected from the Cancer Cell Line Encyclopedia (CCLE) using the ExperimentHub (v. 2.8.1) and depmap (v. 1.14.0) packages in R (v. 4.3.1) based on the 22Q2 dataset (accession number: EH7556). Thirteen cell lines were chosen: A549, PC9, H596, H2228, H358, H1975, H1666, A427, H3122, Calu3, H1650, H1568, and HCC827. The TPM values of each gene were converted to log2(TPM + 1) values and correlated to the *PD-L1v4*/*PD-L1v1* mRNA ratio. The H1993 cell line was not included in this analysis because no gene expression data were available in the 22Q2 dataset from this cell line. RNA processing genes were identified as genes associated with the Gene Ontology (GO) term “RNA processing” (GO:0006396) or its daughter terms (*n* = 288 GO terms) using the topGO (v. 2.52.0) and GO.db (v. 3.17.0) packages in R. This resulted in the identification of 923 genes associated with RNA processing.

RNA sequencing-based expression data for NSCLC in vivo material were extracted from TCGA using the MapLab tool Wanderer (TCGA Wanderer: An interactive viewer to explore DNA methylation and gene expression data in human cancer (http://maplab.imppc.org/wanderer/) (accessed on 1 August 2023). We examined the lung adenocarcinoma (LUAD) and lung squamous carcinoma (LUSC) datasets, as well as the datasets including expression data for corresponding normal lung tissue. Unpaired Student’s *t*-tests, Spearman correlation analyses, and Pearson correlation analyses were performed on the expression data using the resource http://vasserstats.net (accessed on 1 September 2023). The experimentally determined expression data are presented as means and error bars as standard deviation. Significance was analyzed using Student’s *t*-tests, a one-way ANOVA with a Holm–Šídák multiple comparisons test, or a two-way ANOVA with a Holm–Šídák multiple comparisons test. Data were considered significant when *p*-value/adjusted *p*-value < 0.05.

## 3. Results

### 3.1. PD-L1 mRNA Variants in NSCLC Cell Lines

The alternative *PD-L1* mRNA processing, which results in the synthesis of either *PD-L1v1* mRNA of seven exons encoding the reference mPD-L1 protein or *PD-L1v4* mRNA of four exons encoding the sPD-L1 protein, is illustrated in Figure 1A. *PD-L1v4* mRNA is a result of the lack of use of the intron 4 (I4) 5′-splice site (5′SS) together with the usage of an I4 poly-A signal and with the most commonly used I4 poly-A signal being located 95bp downstream the 5’SS (Figure 1A). Thereby, exon 4 (E4) is elongated, and the resulting exon 4A (E4A) will be the terminal exon in the alternative transcript (Figure 1A).

This difference in transcript-included sequences allowed for the specific detection of each *PD-L1* variant from cDNA using RT-qPCR and ddPCR analyses, and the two methods gave concordant results for expression (Appendix A). We addressed the levels of the mRNA variants *PD-L1v1* and *PD-L1v4* in a collection of 14 NSCLC cell lines. The result illustrated a co-expression of the variants in alignment with the two variants being transcriptionally initiated from the same promoter. However, their expression did not significantly correlate, suggesting the participation of other regulatory mechanisms in the determination of the relative transcript ratio (Figure 1B). We next calculated the relative expression ratio between *PD-L1v4* mRNA and *PD-L1v1* mRNA and performed a correlation analysis between the total *PD-L1* mRNA expression and the *PD-L1v4*/*PD-L1v1* mRNA ratio (Figure 1C and Appendix A). This showed that, among the group of examined NSCLC cell lines, such an inverse correlation was not significant (Figure 1C). Thus, the efficiency of transcriptional initiation from the *PD-L1* promoter seems not to be the major determinant for the relative ratio between *PD-L1v4* mRNA and *PD-L1v1* mRNA.

It was previously shown in cancer cell lines that the IFN-γ-mediated induction of *PD-L1* mRNA expression affected *PD-L1v1* mRNA and *PD-L1v4* mRNA expression levels differently, and accordingly, IFN-γ could influence the *PD-L1v4*/*PD-L1v1* mRNA ratio [36,37]. We examined the consequence of IFN-γ stimulation for the *PD-L1v4*/*PD-L1v1* mRNA ratio in three NSCLC cell lines. HCC827 harbors an oncogenic EGFR E746-A750 deletion, and H358 and A549 harbor G12C and G12S KRAS missense mutations, respectively. Of the three cell lines, the total *PD-L1* expression was lowest in A549 cells and highest in H358 cells (Figure 1C), and the transcriptional induction of total *PD-L1* mRNA as a consequence of IFN-γ stimulation was highest in the cells with the lowest total *PD-L1* mRNA expression level (Figure 1D). Moreover, for the three cell lines, there was a decrease in the *PD-L1v4*/*PD-L1v1* mRNA ratio as a consequence of IFN-γ stimulation (Figure 1D). This was a result of a more pronounced induction of *PD-L1v1* mRNA expression relative to *PD-L1v4* mRNA expression (Figure 1D). Altogether, the expression analyses indicate that the *PD-L1v4*/*PD-L1v1* mRNA ratio in the examined NSCLC cell lines is defined by a dynamic interaction between the *cis*-sequences in the *PD-L1* gene and RNA regulatory trans-factors.

### 3.2. In Silico Identification of Trans-Regulators of the PD-L1 Transcript Ratio

We next focused on regulatory mechanisms defining the *PD-L1v4*/*PD-L1v1* mRNA ratio. The mRNA *cis*-sequences required for the generation of the *PD-L1v4* mRNA variant are relatively well-described. However, limited knowledge exists concerning the trans-factors involved in the regulation of the *PD-L1v4*/*PD-L1v1* mRNA ratio and, subsequently, determining the sPD-L1 and mPD-L1 balance. We hypothesized that RNA processing factors that have an expression level in NSCLC cells that correlates, positively or negatively, with the *PD-L1v4*/*PD-L1v1* mRNA ratio could be candidate trans-factors. To identify such candidate trans-factors in silico, we performed a correlation analysis between the RNA-seq-determined expression (TPM_22Q2) of genes in the CCLE subset NSCLC cell lines (A549, PC9, H596, H2228, H358, H1975, H1666, A427, H3122, Calu3, H1650, H1568, and HCC827, *n* = 13) and the PCR-determined *PD-L1v4*/*PD-L1v1* mRNA ratio. The TPM values of each gene were converted to log2(TPM + 1) values and correlated to the *PD-L1v4*/*PD-L1v1* mRNA ratio. Genes with a Spearman correlation coefficient (r) < −0.5 or >0.5 were considered candidate genes. This resulted in the identification of 1035 genes (Figure 2A). These candidate genes were next filtered for being assigned to have a function in RNA processing (GO:0006396) or its daughter terms, yielding 288 candidate GO terms, with 923 genes assigned. This resulted in a final list of 34 candidate genes, with 15 genes having an r < −0.5 and 19 genes having an r > 0.5. These genes represent a potential transcriptomic signature for the *PD-L1v4*/*PD-L1v1* mRNA ratio and, accordingly, the balance between sPD-L1 and mPD-L1 in NSCLC cells (Figure 2A,B, Appendix A).

### 3.3. Analysis of the Trans-Factors CLK2, KHDRBS3, and CDK12

We next examined in vitro three genes from the list of 34 candidate trans-regulators for the *PD-L1v4*/*PD-L1v1* mRNA ratio. We focused on the effect of decreasing the activity of *CDK12, CLK2*, and *KHDRBS3,* which all presented negative Spearman correlations with the *PD-L1v4*/*PD-L1v1* mRNA ratio, but to different extents (*CDK12:* r = −0.746, *CLK2:* r = −0.638, and *KHDRBS3:* r = −0.512) (Figure 2B and Appendix A). This examination specifically addressed genes with a negative correlation, as we anticipated that decreasing the activity of genes presented with a positive correlation would further decrease the low *PD-L1v4* mRNA expression to a level incompatible with reproducible quantification. With CRISPR/Cas9 and specific sgRNAs for each of the three genes, we generated HCC827 cell populations with *CDK12* (two sgRNAs), *CLK2* (two sgRNAs), and *KHDRBS3* (three sgRNAs) indels (Appendix A). For *CDK12* and *CLK2,* the knock-out (KO) scores were not above approximately 50%. These two genes are scored to be common essential genes in pan-cancer cell lines (DepMap Public 23Q2), supporting that there could be a selection for the maintenance of at least one functional allele. This residual expression should be taken into account in the subsequent analyses. For *KHDRBS3,* the KO efficiency was higher, and this gene was not scored to be essential in pan-cancer cell lines (DepMap Public 23Q2). All three sgRNA-targeted genes had lowered mRNA expression following the genetic modifications (Appendix A). We note that the difference in their expression between the control and specific sgRNA cell lines is within the observed expression range of these factors among the 13 NSCLC cell lines where they were identified according to their negative correlation to the *PD-L1v4*/*PD-L1v1* mRNA ratio (Appendix A). HCC827 cells harboring *KHDRBS3* and *CLK2* indels did not show the expected increase in the *PD-L1v4*/*PD-L1v1* mRNA ratio (Figure 2C). However, for the HCC827 cell populations harboring CRISPR/Cas9-mediated indels, we observed an increased expression of the total amount of *PD-L1* mRNA relative to the expression in the HCC827 control cells harboring scrambled sgRNA (Appendix A). We note that the correlation analyses used to identify trans-factors could also identify regulators of the total *PD-L1* mRNA level across the examined cell lines, given the tendency of a negative correlation between the *PD-L1v4*/*PD-L1v1* mRNA ratio and the total *PD-L1* mRNA expression level (Figure 1C). However, analyzing NSCLC CCLE mRNA expression data (Dependency Map portal, *n* = 137) revealed only an insignificant correlation between total *PD-L1* mRNA expression and *CLK2* mRNA expression (Spearman: r = −0.07, *p* = 0.71), as well as between total *PD-L1* mRNA expression and *KHDRBS3* mRNA expression (r = 0.01, *p* = 0.99) (Appendix A). For the HCC827 cells harboring *CDK12* indels, there was a two-fold increase in the *PD-L1v4*/*PD-L1v1* mRNA ratio relative to the control HCC827 cells (Figure 2D and Appendix A). Notably, among the 137 cell lines in the NSCLC CCLE, there was a weak negative correlation between *CDK12* mRNA and total *PD-L1* mRNA expression (Spearman: r = −0.17, *p* = 0.014) (Appendix A). The total *PD-L1* mRNA expression varied considerably between the examined NSCLC cell lines (approximately 20-fold), whereas the variance in *CDK12* mRNA expression was less pronounced (approximately three-fold) (Appendix A). Thus, the *CDK12* indel fraction reflects well the natural variance in *CDK12* mRNA expression and, more importantly, *CDK12* indels have a functional consequence for the *PD-L1v4*/*PD-L1v1* mRNA ratio. Addressing the *PD-L1v1* mRNA and *PD-L1v4* mRNA expression levels as a consequence of *CDK12* indels showed that this primarily results in increased *PD-L1v4* mRNA expression (Figure 2D and Appendix A). The Western blot for mPD-L1 did not reveal major expression differences as a result of the achieved *CDK12* indel efficiency, and the ELISA showed a tendency for increased sPD-L1 expression (Figure 2E). We next addressed the consequence of the functional reduction in CDK12 obtained from CRISPR/Cas9 indels for IFN-γ-induced *PD-L1* mRNA expression. This showed that the increase in the *PD-L1v4*/*PD-L1v1* mRNA ratio in HCC827 cells with *CDK12* indels, relative to sgRNA control cells seen under non-stimulatory conditions, was maintained upon IFN-γ stimulation (Figure 2F). In alignment with *CDK12* indels resulting in a relative increase in *PD-L1v4* mRNA expression, both before and after IFN-γ stimulation (Figure 2D and Appendix A), there was a concordant tendency for a relative increase in sPD-L1 expression, both before and after IFN-γ stimulation (Figure 2F). We conclude that the IFN-γ-mediated stimulation of *PD-L1* transcription, with the resulting relative increase in *PD-L1v1* mRNA compared to *PD-L1v4* mRNA, is not influenced by the reduction in CDK12 function obtained at the given *CDK12* indel frequency. For *PD-L2* mRNA, the presence of *CDK12* indels had no major effect on expression, either before or after stimulation with IFN-γ for 24 h (Appendix A).

### 3.4. THZ531-Mediated Regulation of the PD-L1v4/PD-L1v1 mRNA Ratio

Given the obtained correlation and functional data for CDK12, we focused on this potential trans-regulator of the *PD-L1v4*/*PD-L1v1* mRNA ratio in further analyses. CDK12 has RNA polymerase II (POLII) C-terminal domain (CTD) kinase activity and mediates POLII elongation, processivity, and the read-through of the premature polyadenylation sites located in introns [44,45]. CDK12 and the homologous CDK13 share the same cognate cyclin, cyclin K, and display substantial functional redundancy, except for the specific function of CDK12 to suppress the usage of intronic poly-A sites [46,47]. We note that the *PD-L1v4*/*PD-L1v1* mRNA ratio did not correlate with *CDK13* mRNA expression in the examined NSCLC CCLE cohort (Appendix A), and that no correlation exists between *CDK13* mRNA and *PD-L1* mRNA expression in NSCLC cell lines (Spearman: r = −0.02, *p* = 0.64) (Appendix A). CDK12 can be therapeutically targeted with the covalent inhibitor THZ531, which also targets CDK13 [48]. We tested the effect of THZ531 supplementation on the *PD-L1v4*/*PD-L1v1* mRNA ratio in HCC827 cells. Various concentrations (50 nM, 100 nM, and 200 nM) of THZ531 increased the *PD-L1v4*/*PD-L1v1* mRNA ratio (Figure 3A and Appendix A). Concentrations of THZ531 higher than 200 nM resulted in massive cell death in alignment with THZ531 having an apoptosis-inducing effect at high doses (Appendix A) [48]. The THZ531-mediated increase in the *PD-L1v4*/*PD-L1v1* mRNA ratio was a consequence of decreased *PD-L1v1* mRNA expression. Therefore, THZ531 does not directly imitate the effect of *CDK12* indels which increase the *PD-L1v4*/*PD-L1v1* mRNA ratio through increased *PD-L1v4* mRNA expression. At the protein level, 100 nM of THZ531 did not impair the expression of sPD-L1, whereas a detectable minor decrease in mPD-L1 expression was comparable to the effect of THZ531 on *PD-L1v1* expression (Figure 3A). We next combined THZ531 treatment and IFN-γ stimulation to examine the combined effect in HCC827 cells. Again, at a concentration of 100 nM, THZ531 increased the *PD-L1v4*/*PD-L1v1* mRNA ratio (Figure 3B). Interestingly, THZ531 very efficiently inhibited the IFN-γ-stimulation-mediated induction of *PD-L1* mRNA expression and conferred a *PD-L1v4*/*PD-L1v1* mRNA ratio similar to that observed with only the addition of THZ531 (Figure 3B). To verify that the THZ531 effect for *PD-L1* mRNA expression was not specific for HCC827 cells, we examined the effect of combined treatment of H358 and A549 cells and obtained concordant results (Figure 3C,D). For *PD-L2* mRNA expression, a THZ531 supplement resulted in a decrease and abolishment of an IFN-γ-stimulated mRNA induction (Appendix A). We conclude that supplementation with THZ531 and the resulting inhibition of CDK12 and CDK13 activity abolish IFN-γ-induced *PD-L1* mRNA expression.

### 3.5. PD-L1 Expression Analyses in NSCLC Tumors and Corresponding Normal Lung Tissue

As a proof of principle for the presence of *PD-L1* mRNA variants in an NSCLC tumor context, we first analyzed RNA extracted from FFPE tissue slides for a small cohort of NSCLC adenocarcinoma tumors. For 14 tumors, we measured *PD-L1* mRNA expression significantly above the detection threshold, and in this cohort, there was a *PD-L1v4*/*PD-L1v1* mRNA ratio comparable to the observations in NSCLC cell lines, as well as the presence of a negative correlation between the *PD-L1v4*/*PD-L1v1* mRNA ratio and total *PD-L1* mRNA expression (Figure 4A). These data are in line with other studies addressing the *PD-L1v4* transcript in tumor samples [34,35,36,37,38]. Acknowledging that we only have access to a limited number of tumor samples and therefore low statistical power, we elaborated the initial proof-of-principle analysis using mRNA expression data from TCGA. This allowed an examination of the LUAD and LUSC datasets, which contain expression data of 488 and 491 tumors, respectively, as well as an examination of corresponding normal tissue samples from 58 and 50 individuals, respectively. In both LUAD and LUSC, the total *PD-L1* mRNA expression was decreased, whereas *CDK12* mRNA and *CDK13* mRNA expression was increased, relative to normal lung tissue (Appendix A). The tumor tissue also revealed an increase in *CLK2* mRNA expression, whereas the *KHDRBS3* mRNA expression was lower, compared to normal tissue (Appendix A). For normal tissue corresponding to the LUAD and LUSC samples, the expression correlation analyses showed a correlation between total *PD-L1* mRNA and *CDK12* mRNA expression (r = 0.48 and r = 0.38, respectively) (Appendix A). The correlation was insignificant between *CDK13* mRNA and total *PD-L1* mRNA expression, whereas *CLK2* mRNA and the total *PD-L1* mRNA expression inversely correlated (Appendix A). This was particularly true for the normal tissue corresponding to LUSC (Appendix A). In LUAD and LUSC tumors, the correlation between *CDK12* mRNA and total *PD-L1* mRNA expression was weaker, and, as observed for normal tissue, the expression correlation between *CLK2* mRNA and total *PD-L1* mRNA was negative (Appendix A).

From the TCGA mRNA expression data, we extracted *PD-L1* TPM values for E4 (chr9:5462834–5463121) as well as E6 (chr9:5466770–5466829). Thus, the TPM values for E6 represent *PD-L1v1* mRNA, the TPM values for E4 represent *PD-L1v1* mRNA and *PD-L1v4* mRNA, and the E4/E6 mRNA ratio reflects the relative elongation over the poly-A signals located in I4. We note that the TPM values for E6 could be larger than for E4. This was in alignment with an observed increase in TPM values for downstream exons of genes in the examined TCGA datasets. Both in the LUAD and LUSC datasets, the E4/E6 mRNA ratio was increased in the tumor tissue relative to normal tissue (Figure 4B). Thus, the decrease in total *PD-L1* mRNA expression in NSCLC tumor tissue versus normal tissue is associated with a decreased read-through of the *PD-L1* I4 poly-A sites. Notably, the increase in the E4/E6 mRNA ratio in tumor versus normal tissue was also associated with increased *CDK12* mRNA expression (Appendix A).

Next, we addressed the association between the read-through of the *PD-L1* I4 poly-A sites and the expression of different potential trans-factors in tumor and normal tissue samples. We used Pearson correlation analyses, given that several of the tumor samples possessing E4 sequence reads had no reads for E6, and consequently, impaired conditions for the use of rank sorting in the correlation analyses. In LUAD tumors, the E4/E6 mRNA ratio correlated negatively with the total expression of *PD-L1* mRNA (the sum of the expression for all exons) (r = −0.47) (Figure 4C). This followed the observations from our tumor sample cohort (*n* = 14, Figure 4A) and the result of comparing normal and tumor NSCLC tissues. *CDK12* mRNA expression presented a modest (r = −0.19) negative correlation to the E4/E6 mRNA ratio (Figure 4C). The mRNA expression of none of the other examined factors correlated with the E4/E6 mRNA ratio (Figure 4C). In the corresponding normal tissue, we observed no correlation between the expression of examined factors and the E4/E6 mRNA ratio and, accordingly, the read-through of the *PD-L1* I4 poly-A sites (Figure 4C). In the LUSC tumors and corresponding normal tissue samples, the results were, in general, similar to the observations in the LUAD tumors and corresponding normal tissue. Most notably, the negative correlation between the E4/E6 mRNA ratio and *CDK12* mRNA expression was insignificant in the LUSC tumors. Instead, the LUSC tumors presented a significant negative correlation between the E4/E6 mRNA ratio and *CKD13* mRNA expression (Figure 4C). This is in line with previous observations that *PD-L1* mRNA expression, at least to some extent, is regulated differently in LUAD and LUSC [49].

The total *PD-L1* mRNA expression varied considerably among the tumor samples for both LUAD and LUSC (Appendix A). To take this variability into account, we next addressed subsets of NSCLC tumors with high (50 tumors) and low (50 tumors) *PD-L1* mRNA expression, respectively. For both LUAD and LUSC tumors, low *PD-L1* mRNA expression was associated with an increased E4/E6 mRNA ratio and, accordingly, an impaired read-through of the *PD-L1* I4 poly-A sites (Figure 4D). This resembled the observations for the LUAD and LUSC tumors versus normal tissue samples, with normal tissue samples having relatively increased *PD-L1* mRNA expression and a decreased E4/E6 mRNA ratio (Figure 4B). However, in contrast to the tumor versus normal tissue comparisons, *CDK12* mRNA expression was reduced in the tumor cell populations with low *PD-L1* mRNA expression and a high E4/E6 mRNA ratio compared to the tumor populations with high *PD-L1* mRNA expression and a low E4/E6 mRNA ratio, and this was particularly evident for LUAD (Figure 4E). For *CDK13* mRNA, an expression reduction was not clear (Figure 4E). We noticed an increased *CLK2* mRNA expression in tumor subpopulations with low *PD-L1* mRNA expression (Figure 4E). Next, an examination of gene expression correlation showed a significant negative correlation (r = −0.47) between the E4/E6 mRNA ratio and *CDK12* mRNA expression for LUAD tumors with low *PD-L1* mRNA expression (Figure 4F). A significant negative correlation (r = −0.4) between *CDK13* mRNA expression and the E4/E6 mRNA ratio was present in LUSC tumors with high *PD-L1* mRNA expression (Figure 4F). This indicates that LUSC tumors with high *PD-L1* mRNA expression drive the negative correlation observed between the E4/E6 mRNA ratio and *CDK13* mRNA expression in the group of LUSC tumor samples (Figure 4C).

## 4. Discussion

With this study, we aimed to initiate the identification of *trans*-factors that define the ratio between *PD-L1v4* mRNA and *PD-L1v1* mRNA, and accordingly control the balance between sPD-L1 and exoPD-L1/mPD-L1. The *cis*-sequences determining the generation of *PD-L1v4* mRNA are relatively well-described. *PD-L1v4* mRNA is a result of the skipped use of the canonical *PD-L1* exon 4 donor splice site, together with the use of an otherwise skipped polyadenylation site contributed from a LINE *L2A* element located in I4 or other further downstream I4-located poly-A sites [34,35,36]. The *L2A* insertion in I4 is conserved among mammalians. However, in rodents, a major divergence in the *L2A* sequence, in combination with an alteration in the sequence of the I4 5′-SS, dictates the lack of a rodent *PD-L1v4* mRNA variant [36]. Other mRNA *cis*-sequence elements determining the generation of the *PD-L1v4* mRNA variant have also been identified. These include the presence of weak I4 and I5 5′-SS strength (a score of 85% and 76%, respectively) compared to optimal splice sites (a score of ≥92%), as well as other genetic alterations in I4, including recurrent human papillomavirus integration [34]. To identify trans-factors cooperating with the *cis*-sequences in regulating the *PD-L1v4*/*PD-L1v1* mRNA ratio in NSCLC cells, we performed in silico analyses followed by experimental analyses. We hereby identified CDK12 as a trans-factor with an impact on the *PD-L1* mRNA variant ratio. CDK12, like the homologous CDK13, supports POLII transcriptional elongation by phosphorylating the CTD of POLII after the transcriptional initiation-associated CTD modifications mediated by, e.g., CDK7 and CDK9 [46,47]. A decrease in CDK12 activity can result in an impaired POLII transcriptional elongation rate over intronic sequences, which increases the window of opportunity for cleavage and polyadenylation complexes to process intron-localized poly-A sites. This allows the generation of an alternative 3‘-end sequence in the resulting mRNA [44,45,46,47]. The function of CDK12 to minimize premature termination and polyadenylation at intronic polyadenylation sites is, in particular, observed for long genes involved in DNA damage responses (DDRs) and homologous recombination (HR), e.g., *BRCA1*, *BARD1*, *RAD51*, and *ATR* [44,46,47,50]. CDK13 cannot phenocopy CDK12 in this function [46,47]. The transcriptional regulation of long DDR and HR genes by CDK12 is associated with a relatively low ratio of U1 snRNP binding to intronic polyadenylation sites and, accordingly, less strong U1 telescripting [47,51]. The characterized function of CDK12 to support intronic poly-A site read-throughs for long DDR and HR genes is in alignment with the hereby presented observations for CDK12 to support the read-through of the *PD-L1* I4 poly-A sites to confer *PD-L1v1* mRNA expression, and thereby lowering the *PD-L1v4/PD-L1v1* mRNA ratio. We identified this function of CDK12 in NSCLC cell lines which display a strong negative correlation between *CDK12* mRNA expression and the *PD-L1v4/PD-L1v1* mRNA ratio. To address if a similar correlation was present in NSCLC tumor samples, we performed an expression analysis based on mRNA expression data from the TCGA datasets of LUAD and LUSC. As a measure of the amount of *PD-L1* transcripts that terminate at the I4 poly-A sites, we estimated the ratio of *PD-L1* mRNA transcripts that include E4 relative to *PD-L1* mRNA transcripts that include E6. One putative limitation of this approach could be the existence of additional *PD-L1* mRNA variants besides *PD-L1v1* and *PD-L1v4,* which could interfere with the interpretation. To the best of our knowledge, the existence of E4 skipping is not described in genomic databases and the literature, but we note the existence of a *PD-L1* transcript, *PD-L1-lnc*, which excludes both a part of the E4 sequence and the E5 and E6 sequences [52]. *PD-L1-lnc* is expressed in LUAD and LUSC tumors [52]. However, since this non-coding RNA still includes a large part of the *PD-L1* E4, we anticipate it to have no major impact on the given analyses. In our analyses of LUAD and LUSC samples, we identified only a minor negative correlation between *CDK12* mRNA expression and the E4/E6 mRNA ratio. This result differed from the result derived from the NSCLC cell lines. However, when we grouped the NSCLC tumors relative to high and low *PD-L1* mRNA expression, a striking negative correlation was evident between *CDK12* mRNA expression and the E4/E6 mRNA ratio in the *PD-L1* mRNA low-expressing group, and this was specific for the LUAD tumors. In our initial studies identifying *CDK12* mRNA expression to be negatively correlated with the *PD-L1v4/PD-L1v1* mRNA ratio, we examined NSCLC cell lines of adenocarcinoma origin. These NSCLC cell lines have low *PD-L1* mRNA expression compared to the median *PD-L1* mRNA expression in the tumor samples. Thus, our cell line data are coherent with the observations in the LUAD tumor cohort, since the NSCLC cell lines best resemble the LUAD samples with low *PD-L1* mRNA expression. The negative correlation between *CDK12* mRNA expression and the E4/E6 mRNA ratio was insignificant in LUSC samples with low *PD-L1* mRNA expression and in both LUSC and LUAD samples presented with high *PD-L1* mRNA expression.

It can be envisaged that the regulation of the *PD-L1v4*/*PD-L1v1* mRNA ratio is not only mediated by CDK12, but that the mRNA processing phenotype resulting from low CDK12 expression also can be mimicked through the altered expression of other trans-factors involved in transcriptional elongation and RNA splicing, cleavage, and polyadenylation. It is important to stress the enormous complexity, including multiple regulatory factors that coordinate the co-transcriptional events that influence poly-A signal selection [53]. We note the hereby identified list of 34 potential mRNA regulatory factors whose expression correlated with *PD-L1* I4 poly-A site processing in NSCLC cell lines. Of the two trans-factors we selected for characterization besides CDK12, CLK2 also possesses a function in transcriptional elongation. CLK2 interacts with CDK8, which is a component of the Mediator complex required for POLII transcription, and CLK2, together with CDK8, links cell metabolism status through the mammalian target of the rapamycin (mTOR) pathway, with alternative poly-A site selection [54,55]. We notice an increased *CLK2* mRNA expression in LUAD and LUSC tumors compared to normal tissue, and that *CLK2* mRNA expression is elevated in tumor subpopulations with low *PD-L1* mRNA expression relative to high expression. Notably, although our analyses did not support CLK2 to have an impact on the *PD-L1v4*/*PD-L1v1* mRNA ratio, we identified an impact on the total *PD-L1* mRNA expression. Further in this line, we observed that the IFN-γ-mediated stimulation of *PD-L1* transcription, and the resulting relative increase in *PD-L1v1* mRNA compared to *PD-L1v4* mRNA, is not influenced by the presence of *CDK12* indels. This is in alignment with previous results showing that IFN-γ stimulation results in the assembly of a novel type of transcriptional initiator complex at the *PD-L1* promoter, and that a stimulatory effect of IFN-γ for transcriptional activity is governed by POLII pause release in a process involving CDK8 and CDK19 together with the Mediator complex [20,56].

CDK12 is involved in tumorigenesis and is assigned oncogenic functions in, e.g., HER2-positive breast cancer, as well as tumor suppressor functions in, e.g., ovarian, prostate, and triple-negative breast cancer [44,45,47,50]. The oncogenic functions of CDK12 are proposed to reflect the transcriptional stimulation of, e.g., *MYC* and *EWS/FLI* gene expression, while tumor-suppressing functions are proposed to be a consequence of stimulating the expression of, in particular, long genes in the DDR and HR pathways [44,45,47,50]. Impairment of the DDR and HR pathways results in genomic instability; generates neoantigens; interferes with IFN signaling and cyclic GMP-AMP synthase–stimulator of interferon genes (cGAS-STING) signaling; and upregulates PD-L1 expression [57]. In this line, *CDK12* loss-of-function mutations in ovarian cancer cells sensitize the cancer cells to the effect of poly ADP-ribose polymerase (PARP) and CHK1 inhibitors. This synthetic lethality phenocopies the effect of *BRCA1* mutations and is a consequence of CDK12 being an upstream factor in the BRCA1 DNA repair pathway [44]. Similarly, *CDK12* loss-of-function mutations in prostate cancer sensitizes the cancer cells to androgen receptor antagonists [44,58,59]. The contribution of CDK12 to NSCLC tumorigenesis is relatively undescribed, but genomic alterations in the *CDK12* gene occur in approximately 5% of the tumors [50]. We identified that *CDK12* is more expressed in LUAD and LUSC tumor tissue relative to normal tissue, and this is in alignment with a recent study by Liu et al. [60]. Interestingly, *CDK12* mutations in NSCLC samples did not impact the expression of a small subset of examined DDR and HR pathway genes [60]. Moreover, the siRNA-mediated depletion of CDK12 expression in the NSCLC cell lines A549 and H1299, harboring *KRAS* and *NRAS* mutations, respectively, conferred the inhibition of cell growth, as well as the growth using in vivo models, by inducing apoptosis [60]. The effect of CDK12 depletion was proposed to be mediated by the downregulation of the expression of TBK1, which is encoded from a relatively long gene (100 kb). The study by Liu et al. indicates that CDK12 is a potential novel treatment target of NSCLC [60]. This notion is particularly important given that the function of CDK12 can be inhibited by small-molecule inhibitors that have already been explored related to, e.g., breast, ovarian, and prostate cancer therapy [50,61,62]. THZ531 is an example of a small-molecule covalent inhibitor of CDK12 and CDK13 activity, with half-maximal inhibitory concentrations of 158 nM and 69 nM, respectively [48]. The inhibitory effect of THZ531 for the related CDKs, CDK7 and CDK9, is more than 50-fold weaker, and additional off-target effects are not well described [48]. Notably, CDK12 also phosphorylates gene regulatory factors beyond the POLII CTD, and is also involved in other cellular processes, e.g., translational regulation [44]. Thus, interference with CDK12 activity with THZ531, as well as CDK12 overexpression and depletion analyses, can have effects beyond an effect at the transcriptional level through POLII CTD phosphorylation. From a cancer therapeutic perspective, it is shown that THZ531 triggers apoptosis in cancer cells, and THZ531 is accordingly a promising stand-alone therapy in, e.g., breast cancer and hepatocellular carcinoma [48,62]. This could also be relevant for NSCLC. Moreover, since THZ531 phenocopies the effect of *CDK12* loss-of-function mutations to mediate the downregulation of DDR and HR pathway genes, it is also proposed to combine CDK12 inhibition with, e.g., PARP inhibitors, CHK1 inhibitors, tyrosine kinase inhibitors, and chemotherapeutic drugs to mediate synthetic lethality [46,48,62,63,64]. Whether the latter could be meaningful for NSCLC is still in its infancy, but it could be promising that synthetic lethality is described between CDK12 inhibition and EGFR tyrosine kinase inhibitors, since EGFR deregulation is a recurrently observed cancer-driving event in NSCLC [1,63,64].

The hereby presented observation that CDK12 regulates *PD-L1* mRNA variant expression and helps to dictate the balance of sPD-L1 relative to mPD-L1/exoPD-L1 brings CDK12 and CDK12 inhibitors into a novel immunotherapeutic context with relevance for NSCLC and other cancer types. We observed that the effect of THZ531 to increase the *PD-L1v4*/*PD-L1v1* mRNA ratio was a consequence of decreased *PD-L1v1* mRNA expression, and is thereby different than the effect of *CDK12* indels which mediated increased *PD-L1v4* mRNA expression. Since THZ531 inhibits both CDK12 and CDK13, this difference could be a consequence of CDK13 inhibition resulting in decreased transcriptional elongation over the 3′ end of the *PD-L1* gene; meanwhile, the concomitant CDK12 inhibition facilitated the use of the I4 poly-A sites, resulting in a transcriptional negative effect, primarily manifested for *PD-L1v1* mRNA. The current literature mostly supports that sPD-L1 is less efficient compared to cancer-cell-surface mPD-L1 in suppressing T cell activation through PD-1 interaction [29,33,34,35,36,37,38]. It is still too early to deduce that increasing the sPD-L1 relative to mPD-L1 expression, e.g., conferred by CDK12 inhibition, will be beneficial for the PD-1/PD-L1-pathway immunotherapeutic response. Related to this, CDK12 and CDK13 inhibition in breast cancer induces immunogenic cell death and enhances PD-1 immunotherapy [65], while on the other hand, defective transcriptional elongation is described to confer immunotherapy resistance for some cancers [66]. Further experimental studies are needed to delineate the immunotherapeutic potential of the regulation of the sPD-L1/mPD-L1 balance through the inhibition of CDK12.

## 5. Conclusions

This study aimed to identify regulatory factors influencing the balance between *PD-L1v4* and *PD-L1v1* mRNA variants and their impact on the balance between sPD-L1 and exoPD-L1/mPD-L1 in NSCLC cells. Through in silico and experimental analyses, we identified that the POLII CTD kinase CDK12 plays a role in regulating the *PD-L1v4/PD-L1v1* mRNA ratio. CDK12 supports the transcriptional bypassing of intron-located poly-A signals, especially for long DDR and HR genes, and this well-characterized CDK12 function is in alignment with the presented observation that decreased CDK12 expression mediates an increased *PD-L1v4/PD-L1v1* mRNA ratio. Altering the sPD-L1 and exoPD-L1/mPD-L1 balance in NSCLC cells through CDK12 inhibition using small molecules like THZ531 might impact the effectiveness of PD-1/PD-L1 immunotherapy, and suggests a potential link to DDR and HR pathway-directed therapy. However, the implications of increasing sPD-L1 relative to exoPD-L1/mPD-L1 expression and its effect on immunotherapeutic responses is still very unclear and requires further experimental investigation, e.g., using patient-derived xenograft and cell-line-derived xenograft translational research models.

## 6. Footnote

During the time of finalization of this manuscript, Zhang et al. published a manuscript with some overlapping conclusions [67]. They showed that the GSK3α/β inhibitors AR-A014418 and THZ531 in some cancer cells (but not including analyses of NSCLC cells) increased the *PD-L1v4*/*PD-L1v1* mRNA ratio by decreasing the activity of CDK12 and CDK13; *CDK12* knockdown promoted *PD-L1* I4 poly-A signal usage; *CDK13* knockdown decreased the mRNA expression of both *PD-L1v1* and *PD-L1v4*; and a co-knockdown of *CDK12* and *CDK13* was required to mimic the *PD-L1* transcriptional effect of THZ531.

## Figures and Tables

**Figure 1 cells-12-02844-f001:**
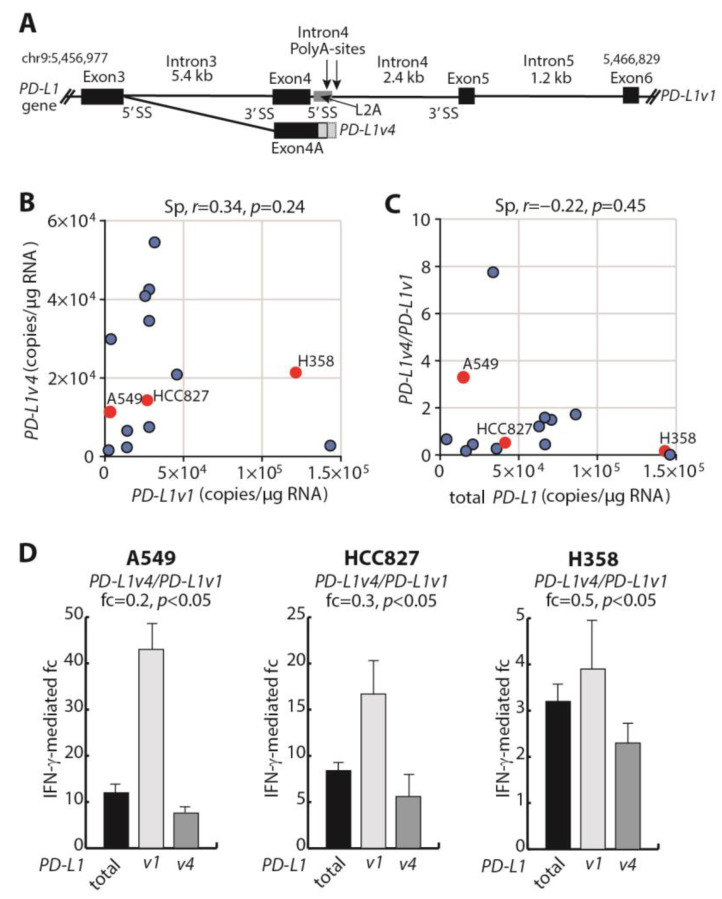
Characterization of alternative exon 4 *PD-L1* pre-mRNA processing. (**A**) Schematic illustration of the region in the *PD-L1* (CD274) gene defining the two mRNAs *PD-L1v1* including exons 1 – 7 and *PD-L1v4* including exons 1 - 3 and 4A. (**B**) ddPCR analysis of the correlation between *PD-L1v1* and *PD-L1v4* mRNA expression measured as copies per μg of RNA in 14 NSCLC cell lines. The Spearman correlation coefficient (r) is given in the upper part. Cell lines A549, HCC827, and H358 are indicated in red. (**C**) Correlation analysis between *PD-L1v4*/*PD-L1v1* mRNA expression ratio and the total expression of *PD-L1* mRNA measured as copies per μg of RNA by ddPCR in 14 NSCLC cell lines. The Spearman correlation coefficient (r) is given in the upper part. Cell lines A549, HCC827, and H358 are indicated in red. (**D**) The fold change (fc) in mRNA expression from stimulation with IFN-γ for 24h in the NSCLC cell lines A549, HCC827, and H358. Expression was determined with RT-qPCR (*n* = 4). The fc for total *PD-L1*, *PD-L1v1*, and *PD-L1v4* mRNA are illustrated with the bar diagrams, and the fc in *PD-L1v4*/*PD-L1v1* mRNA expression ratio is given above for each cell line. Student’s *t*-test was used to examine for significant differences in ratio.

**Figure 2 cells-12-02844-f002:**
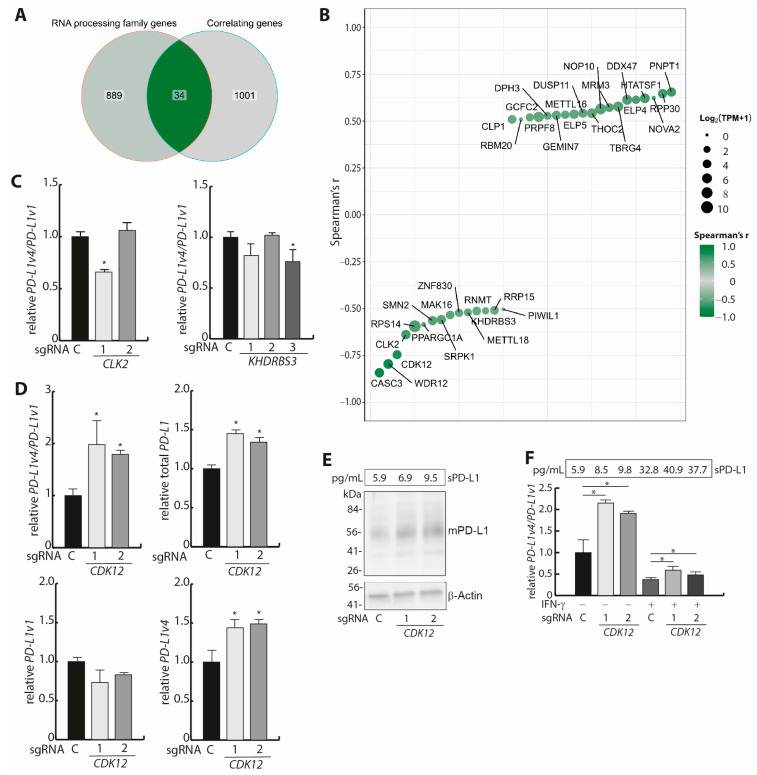
Identification of putative trans-regulators of the *PD-L1v4*/*PD-L1v1* ratio. (**A**) Venn diagram illustrating the overlap between genes assigned to have a function in RNA processing and genes identified to have an expression correlated with the *PD-L1v4*/*PD-L1v1* mRNA ratio in NSCLC cell lines (Spearman correlation coefficient: r < −0.5 or 0.5 < r). (**B**) Illustration of the correlation (Spearman (r), indicated by color) to the *PD-L1v4*/*PD-L1v1* mRNA ratio for the 34 identified putative trans-factors. Size indicates the mean Log2(TPM + 1) across the 13 cell lines. (**C**) The effect of *CLK2* and *KHDRBS3* indels for the *PD-L1v4*/*PD-L1v1* mRNA ratio. CRISPR/Cas9 and specific sgRNAs (two for *CLK2* and three for *KHDRBS3*) were used for indel generation and as a control, c, a scrambled sgRNA. The relative *PD-L1v4*/*PD-L1v1* mRNA ratio was determined using ddPCR data (*n* = 3). Student’s *t*-test was used to examine for significant differences in ratio between targeting sgRNAs and control. * *p* < 0.05. (**D**) The effect of *CDK12* indels for the *PD-L1v4*/*PD-L1v1* mRNA ratio with two specific sgRNAs for *CDK12*. The expression levels were determined using RT-qPCR data normalized to *TBP* (*n* = 2). Student’s *t*-test was used to examine for significant differences in ratio between *CDK12* sgRNAs and control. * *p* < 0.05. (**E**) The effect of *CDK12* indels for sPD-L1 expression measured via enzyme-linked immune absorbent assay (*n* = 3) and for mPD-L1 measured via Western blotting (*n* = 2, a representative is shown). (**F**) The *PD-L1v4*/*PD-L1v1* mRNA ratio was measured with RT-qPCR before and after IFN-γ stimulation for 24h in HCC827 cells (*n* = 2). HCC827 cells with the indicated sgRNA-mediated *CDK12* indels were analyzed. The *PD-L1v4*/*PD-L1v1* mRNA ratio was normalized to the ratio in unstimulated HCC827 cells harboring control sgRNA. The corresponding effects of *CDK12* indels for sPD-L1 expression were measured via enzyme-linked immune absorbent assay (ELISA) (*n* = 3).

**Figure 3 cells-12-02844-f003:**
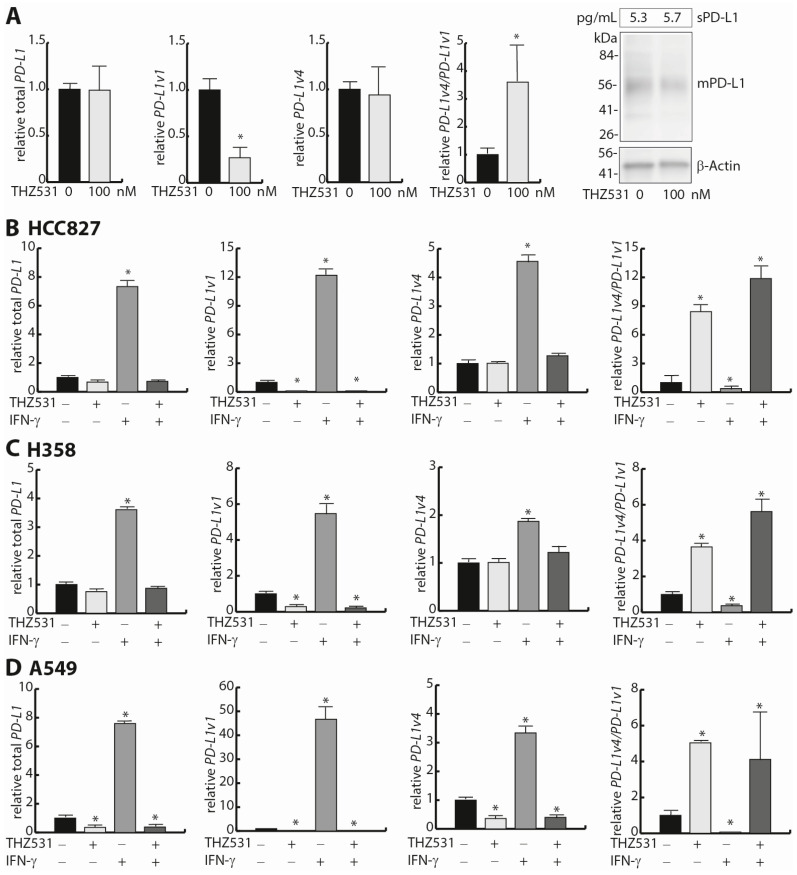
The CDK12/13 inhibitor THZ531 impacts the *PD-L1v4*/*PD-L1v1* ratio. (**A**) The effect of stimulating HCC827 cells with 100 nM of THZ531 for 24 h on expression of *PD-L1* mRNA variants and the *PD-L1v4*/*PD-L1v1* mRNA ratio. Expression was measured with RT-qPCR using *ACTB* as a normalization gene and expression was subsequently normalized to control cells (*n* = 2). The corresponding effects for sPD-L1 expression measured via ELISA (*n* = 3) and for mPD-L1 measured via Western blotting (*n* = 2) are shown to the right. (**B**) The effect of co-stimulation with 100 nM of THZ531 and IFN-γ for 24h on *PD-L1* mRNA expression in HCC827 cells. Expression was measured with RT-qPCR using *ACTB* as a normalization gene and expression was subsequently normalized to control cells (*n* = 2). Student’s *t*-test was used to examine for significant differences in mRNA expression. * *p* < 0.05. (**C**,**D**) as (**B**) but with cell lines H358 and A549, respectively.

**Figure 4 cells-12-02844-f004:**
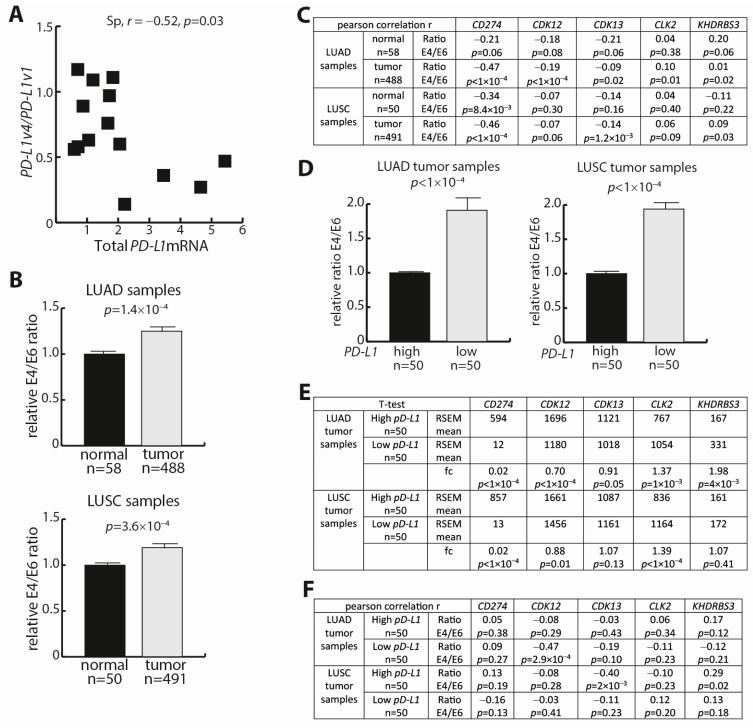
*PD-L1* alternative processing in NSCLC tumors and corresponding normal lung tissue. (**A**) RNA was extracted from formalin-fixed and paraffin-embedded NSCLC tumor samples, and the *PD-L1v4*/*PD-L1v1* mRNA ratio as well as the total *PD-L1* expression were determined via ddPCR (*n* = 14). The Spearman correlation is shown. (**B**) RNA-seq expression data, in format of RNA-seq by expectation-maximization (RSEM), for *PD-L1* mRNA were extracted from The Cancer Genome Atlas (TCGA) subsets lung adenocarcinoma (LUAD) and lung squamous carcinoma (LUSC), as well as the corresponding normal tissue samples. The *PD-L1* E4/E6 mRNA ratios were calculated for the samples. Student’s *t*-test was used to examine for differences in ratio. (**C**) Pearson correlation analyses between E4/E6 mRNA ratio and mRNA expression for the given factors in LUAD, LUSC, and normal samples from TCGA. (**D**) LUAD and LUSC expression data for total *PD-L1* were fractionated into a low-expressing cohort (*n* = 50) and a high-expressing cohort (*n* = 50), and the relative E4/E6 mRNA ratios were calculated. Student’s *t*-test was used to examine for differences in ratio. (**E**) Analyses of mRNA expression levels of the given factors in low and high *PD-L1* mRNA expression cohorts from LUAD and LUSC. Student’s *t*-test was used to examine for differences in expression. (**F**) Pearson correlation analyses of mRNA expression levels of the given factors in low and high *PD-L1* mRNA expression cohorts from LUAD and LUSC.

## Data Availability

All data will be shared upon reasonable request to the corresponding author.

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
