# Peer review of "Trans-Regulation of Alternative PD-L1 mRNA Processing by CDK12 in Non-Small-Cell Lung Cancer Cells"

_cells, 2023, doi:10.3390/cells12242844_

Round 1

Reviewer 1 Report

Comments and Suggestions for Authors

In the manuscript "Trans-Regulation of Alternative PD-L1 mRNA Processing by CDK12 in Non-Small-Cell Lung Cancer Cells" by Larsen et al., the authors aim to identify factors influencing the ratio of soluble to membrane-bound PD-L1 in Non-Small-Cell Lung cancer cells. They determine the  ratio of soluble to membrane-bound PD-L1 in 14 lung cancer cell lines and identify genes based on published expression data for the cell lines that either positively or negatively correlate with the sPD-L1/mPD-L1 ratio. Overall, they identify 34 genes with RNA-interacting properties that correlate with the sPD-L1/mPD-L1 ratio and perform Crispr/Cas9-based knock out on three of these genes. The data show that partial knock down of CDK12 increases the sPD-L1/mPD-L1 ratio due to increased use of the alternative exon 4. They try to verify the results using an CDK12/CDK13 inhibitor as well as published data on LUAD and LUSC samples.

While the hypothesis behind the project is interesting, the data do not show a clear correlation of CDK12 with the sPD-L1/mPD-L1 ratio. While the partial knock down shows an influence, the data are hard to interprete as CDK12 is only deleted in half of the cells and not on both alleles. Further, the expression data obtained from LUAD and LUSC do not support this influence as CDK12 expression is not correlated with the obtained E4/E6 ratio.

Comments:

·         Figure 1B and C: It is not clear, which cell line correspond to which circle in the figure. Should be color-coded.

·         L217: “The result illustrated a tendency for correlation”

o   I do not observe a tendency. This sentence should be changed.

·         Abbreviations have to be spelled out (e.g. TPM)

·         Why did the authors only test genes with a negative correlation and not genes which are positively associated with sPD-L1?

·         L246-261: For me, this is rather introduction than results. Should be moved to the introduction and only shortly summarized in the results section, if necessary

·         Figure S3B: Information are missing, which cell line was used here.

·         L358-377: Again, rather introduction than results.

·         In Figure 2, the authors show that partial know down of CDK12 induces PD-L1v4 expression but does not reduce PD-L1v1 expression. However, when using the inhibitor of CDK12/13 in Figure 3, they have basically the opposite. PD-L1v1 expression is reduced, while PD-L1v4 is not affected. How do the authors explain these differences?

·         Line 432-435: In the screen with the NSCLC cell lines, only correlations with r values above 0.5 or lower than -0.5 were seen as candidate genes. Why now values below 0.5?

·         For the calculation of the PD-L1v1/PD-L1v4 ratio in LUAD and LUSC samples from TCGA, the authors use values for E4 and E6 for the calculation. Can the authors exclude that alternative PD-L1 variants exist that use E6 without E4?

·         Based on the E4/E6 ratio in Figure 4, there is not a strong correlation of CDK12 with the PD-L1v4/v1 ratio. How does this fit to the data obtained with the cell lines?

Comments on the Quality of English Language

I have no comments on the quality of the english language. However, the results part is difficult to follow.

Author Response

We thank the reviewer very much for the very constructive and excellent comments for the improvement of the manuscript. We are very pleased to have the opportunity to address the raised concerns and suggestions for improvement. We have incorporated in the revised manuscript revisions following the comments and suggestions from the reviewer (along with incorporation of revisions to the comments and suggestions from reviewer 2). This has importantly improved the impact and readability of the manuscript. Especially the revised results and discussion sections now presents the results from the study much more straightforward and coherent.  Below is a point-by-point reply to the reviewer's comments and suggestions.

Review report 1:

Comments and Suggestions for Authors

In the manuscript "Trans-Regulation of Alternative PD-L1 mRNA Processing by CDK12 in Non-Small-Cell Lung Cancer Cells" by Larsen et al., the authors aim to identify factors influencing the ratio of soluble to membrane-bound PD-L1 in Non-Small-Cell Lung cancer cells. They determine the  ratio of soluble to membrane-bound PD-L1 in 14 lung cancer cell lines and identify genes based on published expression data for the cell lines that either positively or negatively correlate with the sPD-L1/mPD-L1 ratio. Overall, they identify 34 genes with RNA-interacting properties that correlate with the sPD-L1/mPD-L1 ratio and perform Crispr/Cas9-based knock out on three of these genes. The data show that partial knock down of CDK12 increases the sPD-L1/mPD-L1 ratio due to increased use of the alternative exon 4. They try to verify the results using an CDK12/CDK13 inhibitor as well as published data on LUAD and LUSC samples.

While the hypothesis behind the project is interesting, the data do not show a clear correlation of CDK12 with the sPD-L1/mPD-L1 ratio. While the partial knock down shows an influence, the data are hard to interprete as CDK12 is only deleted in half of the cells and not on both alleles. Further, the expression data obtained from LUAD and LUSC do not support this influence as CDK12 expression is not correlated with the obtained E4/E6 ratio.

Reply 1: We thank the reviewer for the comments. We would like to comment on the last part. We identified CDK12 given the negative correlation to the PD-L1v4/PD-L1v1 mRNA ratio in NSCLC cell lines and verified with knockouts. Indeed we did not obtain a full CDK12 knockout (it is an essential gene in the examined cell lines). However, we obtained knock-out efficiencies of CDK12 which compared to control cells reflected relatively well the normal variance in CDK12 expression in the cell lines. Thus, at least to some extent, was the CDK12 knock-out modeling of the CDK12 expression variance which was hypothesized could confer differences in PD-L1v4/PD-L1v1 mRNA ratio, and a similar effect size was indeed observed. For replication in tumor data sets we note that in a subset of LUAD tumors with low PD-L1 expression (which is a tumor subset mostly resembling the characteristics of the examined NSCLC cell lines) the CDK12 expression correlates with PD-L1 E4/E6 mRNA (pearson r = -0.47 P = 2.9E-4) (Figure 4F). We anticipate this is support of the CDK12 function for PD-L1 isoform generation to also have the potential to be present in a tumor context. These data are in the revised results and discussion sections more clearly described.

Comments:

  • Figure 1B and C: It is not clear, which cell line correspond to which circle in the figure. Should be color-coded. Reply: A549, HCC827, and H358 have been marked with red. Figure legends have been updated.
  • L217: “The result illustrated a tendency for correlation”.I do not observe a tendency. This sentence should be changed. Reply: L202. This correction was made in the revised results section.
  • Abbreviations have to be spelled out (e.g. TPM).
    Reply: LINE, CRISPR, and TPM have been spelled out.
  • Why did the authors only test genes with a negative correlation and not genes which are positively associated with sPD-L1?
    Reply: Additionally, we wanted to validate the candidate genes by knockout. PD-L1v4 expression is in general at the lower end of the expression scale in the NSCLC cell lines and it is lower than PD-L1v1 expression. Thus, it is easier to measure an increase in PD-L1v4 expression than to measure a further decline in PD-L1v4 expression, a decline that will approach the limit of detection by qPCR. By examining candidate genes with a negative correlation, knockout will (assuming that the candidate gene poses a regulatory function as expected from the screen), cause an increase in PD-L1v4 or a decrease in PD-L1v1 expression that we should be able to measure. This argumentation (in a shorter format) to select genes with negative correlation is presented in the revised results section.
  • L246-261: For me, this is rather introduction than results. Should be moved to the introduction and only shortly summarized in the results section, if necessary
    Reply: This was also raised as a concern by reviewer 2, but with the suggestion to move the text to the discussion. We have eliminated most of this text from the results and simply maintained the absolute most necessary information to give background for the results to come. Instead, we have incorporated the text in a revised manner in the discussion. Altogether this has very much improved the readability of the entire manuscript
  • Figure S3B: Information are missing, which cell line was used here.

Reply. HCC827 was used. The information is now given relevant places.

  • L358-377: Again, rather introduction than results.
    Reply. As the concern for L246-261 (see above), this was also a concern for reviewer 2 and we have also eliminated most of this text from the results. Instead, we have incorporated, as suggested by reviewer 2, the text in a revised manner in the discussion. We considered a long introduction to CDK12 in the introduction section difficult to incorporate given identification of CDK12 as a PD-L1 regulator is a main finding in the study. Altogether this has very much improved the readability of the entire manuscript.
  • In Figure 2, the authors show that partial know down of CDK12 induces PD-L1v4 expression but does not reduce PD-L1v1 expression. However, when using the inhibitor of CDK12/13 in Figure 3, they have basically the opposite. PD-L1v1 expression is reduced, while PD-L1v4 is not affected. How do the authors explain these differences?
    Reply. Our best suggestion for this difference is the reduction of PD-L1v1 in Figure 3 must be assigned to the inhibition of CDK13 by THZ531. If both CDK12 and CDK13 are inhibited by THZ531 there will be an in general lower transcriptional elongation over the PD-L1 gene 3’region but since in addition the I4 poly-A site now will be used more the effect will be manifested in particular for the PD-L1v1 mRNA. On the other hand, if only CDK12 is inhibited, e.g. by the knock-out, the effect instead specifically will be that the I4 poly-A site will be used more and therefore more PD-L1v4 mRNA. But since PD-L1v4 mRNA is expressed to a lower level than PD-L1v1 mRNA, the result that more PD-L1v4 is generated on the expenses of the amount of PD-L1v1 mRNA, is not manifested as measurable in the PD-L1v1 amounts. This point is included in the revised discussion.
  • Line 432-435: In the screen with the NSCLC cell lines, only correlations with r values above 0.5 or lower than -0.5 were seen as candidate genes. Why now values below 0.5?
    A threshold of r above 0.5 or lower than -0.5 was used to narrow down the number of candidate genes. Both candidate genes with a negative and positive r are interesting as both can have a trans-regulatory function for the PD-L1 transcript ratio. The reason for validating candidate genes with a negative correlation is described in a previous comment.
  • For the calculation of the PD-L1v1/PD-L1v4 ratio in LUAD and LUSC samples from TCGA, the authors use values for E4 and E6 for the calculation. Can the authors exclude that alternative PD-L1 variants exist that use E6 without E4?
    Reply. We have no indications that variants exist without exon 4. However, variant NCBI NR_052005 has partial skipping of exon 4 as well as exon 5 and 6 and encodes a long non-coding RNA. There could be other variants as well. In our previous very preliminary analyses in NSCLC samples and cell lines, eventual variants excluding exon 4 must most likely be expressed to a level significantly lower than the PD-L1v1 and PD-L1v4 variants. How the existence of PD-L1 variants can impact and limit the presented results is described in the revised discussion section.

Based on the E4/E6 ratio in Figure 4, there is not a strong correlation of CDK12 with the PD-L1v4/v1 ratio. How does this fit to the data obtained with the cell lines?
Reply. (This was also addressed in reply 1 above). For replication of results obtained in NSCLC cell lines in TCGA LUAD and LUSC tumor data sets (Figure 4) we agree that in examinations using the bulk TCGA expression data, the CDK12 correlation was not strong. However, a significant correlation between CDK12 expression and E4/E6 ratio is present among LUAD tumors (r = -0.19 P < 1.0E-4) but not in LUSC tumors. But interestingly, we note if we instead examine a subset of LUAD tumors with low PD-L1 expression (which is a tumor subset mostly resembling the characteristics of the examined NSCLC cell lines) the CDK12 expression correlates with PD-L1 E4/E6 mRNA (pearson r = -0.47 P = 2.9E-4) (Figure 4F). We anticipate this to support the presence of the CDK12 function for PD-L1 isoform generation also existing in a tumor context. These data are in the revised results and discussion sections more clearly described.

On top of this, we note that the TCGA RNA-seq data of tumors represents both cancer cells and in addition other tumor-associated cell types whereas the NSCLC cell line data from the DepMap portal specifically represents cancer cells. A contribution of RNA-seq values from non-cancer cells in the TCGA could also contribute to the observed weaker correlation in LUAD patient samples compared to the NSCLC cell lines. This possible limitation is shortly described in the revised discussion.

Comments on the Quality of English Language

I have no comments on the quality of the english language. However, the results part is difficult to follow.

Reply: We have modified the results section according to the suggestions from reviewer 1 and reviewer 2. Because of this, numerous text elements for introduction and discussion have been removed from the results section. This revision has enormously improved the readability of the results section and improved the impact of the discussion section.

Reviewer 2 Report

Comments and Suggestions for Authors

Authors described the role CDK12 in regulating mPD-L1/sPD-L1. This is an interesting article and could have clinical relevance in NSCLC and would requires preclinical models - CDX and PDX models to validate the hypothesis. 

Comments:

1.The authors state that CDK12 could be considered as an essential gene ( DepMap) How relevant is targeting CDK12 in NSCLC as a standalone therapy or in combination?

2. It would be interesting to visualize the cell viability data with CDK12 inhibitor for Figure 3.

3. Any comment on the off target affects of the drug?

4. Any comments on the CDK12 role in regulating expression of other long protein coding genes that could be important to understand CDK12 mediated signaling. This should be discussed.

Comments on the Quality of English Language

1. The text describing the figures are not matching with the Figure panels. For example - lines 237-239, 339-341. Please check throughout the manuscript. 

2. Authors can cutdown text with citation in result section. Keep this part of the manuscript for discussion. Authors should work on organizing each result section in pieces. Presently test is all over the place. 

3.

Author Response

We thank the reviewer very much for the very constructive and excellent comments for the improvement of the manuscript. We are very pleased to have the opportunity to address the raised concerns and suggestions for improvement. We have incorporated in the revised manuscript revisions following the comments and suggestions from the reviewer (along with incorporation of revisions to the comments and suggestions from reviewer 1). This has importantly improved the impact and readability of the manuscript. The revised results and discussion sections now present the results from the study much more straightforward and coherent.  Below is a point-by-point reply to the reviewer's comments and suggestions

Review report 2:

Comments and Suggestions for Authors

Authors described the role CDK12 in regulating mPD-L1/sPD-L1. This is an interesting article and could have clinical relevance in NSCLC and would requires preclinical models - CDX and PDX models to validate the hypothesis. 

Comments:

1.The authors state that CDK12 could be considered as an essential gene ( DepMap) How relevant is targeting CDK12 in NSCLC as a standalone therapy or in combination?

Reply: We have elaborated on this very interesting question in the revised discussion section and included many more aspects of this compared to the initial discussion. Whereas CDK12 loss or gain of function impacts several cancers (and this is briefly now described in the revised discussion) the impact of CDK12 and CDK12 targeting in NSCLC is relatively sparse. However a novel study by Liu.X. et al DOI: 10.1016/j.mcp.2023.101923) contributes some interesting aspects now included in the revision together with some other relevant studies. We have put this knowledge into the context of our observations to briefly describe, without over-interpretation, the potential of CDK12 targeting in NSCLC either alone, in the context of synthetic lethality, and the context of PD-1/PD-L1 axis immunotherapy. 

  1. It would be interesting to visualize the cell viability data with CDK12 inhibitor for Figure 3.

Reply: That THZ531, especially at high concentrations, has a high apoptosis-inducing effect is convincingly shown in the literature (e.g. Zhang-T et al. Nature Chemical Biology, DOI:10.1038/NCHEBIO.2166). We have in the revised supplemental Figure 5D included an imaging panel showing the effect of various concentrations of THZ531 for HCC837 cell survival.

  1. Any comment on the off target affects of the drug?

Reply: THZ 531 targets CDK12 and CDK13 with IC50 values of approximately 160 nM and 70 nM, respectively. The targeting of the related CDK7 and CDK9 kinases is at least 50-fold lower. Other off targets are to the best of our knowledge not known. This is described in the revised discussion. Moreover, at least CDK12 is proposed to have functions besides POLII CTD phosphorylation inclusive impacting translation. Thus, targeting CDK’s with THZ531 might also target additional cellular pathways than transcriptional elongation-associated processes. This is also now discussed in the revised discussion.

  1. Any comments on the CDK12 role in regulating expression of other long protein coding genes that could be important to understand CDK12 mediated signaling. This should be discussed.

Reply: We have revised the discussion to now include a discussion of this subject but still with a particular focus on long DDR genes. We are confident this extended discussion much better highlights different aspects of CDK12–mediated regulation.

Comments on the Quality of English Language

  1. The text describing the figures are not matching with the Figure panels. For example - lines 237-239, 339-341.
    These inconsistencies are corrected in the revised manuscript.

Please check throughout the manuscript. 
Reply. This is checked throughout the manuscript and eventual inconsistencies are corrected.

  1. Authors can cutdown text with citation in result section. Keep this part of the manuscript for discussion. Authors should work on organizing each result section in pieces. Presently test is all over the place. 

Reply: (This concern was also raised by reviewer 1). We have eliminated most of this specified text from the results and simply maintained the absolute most necessary information to give background for the results to come. Instead, we have now incorporated the text in a revised manner in the discussion along with the other new discussion points recommended by the reviewer. Moreover, some other corrections and text alignment have been performed for the results section. This has very much improved the readability of the results section and at the time to a large extent increased the quality of the discussion and this has altogether improved the impact of the manuscript. 

3.